# Is transcutaneous auricular vagus nerve stimulation effective and safe for primary insomnia? A PRISMA-compliant protocol for a systematic review and meta-analysis

Ting Yang[1☯], Yunhuo Cai[2☯], Xingling Li[3], Lianqiang Fang[3]*, Hantong Hu [3]*

1 Department of Nursing, The Third Affiliated Hospital of Zhejiang Chinese Medical University, Hangzhou City, China, 2 Department of Rehabilitation, The Third Affiliated Hospital of Zhejiang Chinese Medical University, Hangzhou City, China, 3 Department of Acupuncture and Moxibustion, The Third Affiliated Hospital of Zhejiang Chinese Medical University, Hangzhou City, China

☯ These authors contributed equally to this work.
* 413351308@qq.com (HH); fanglianqiang21@163.com (LF)

**Data Availability Statement:** No datasets were generated or analysed during the current study. All

## Abstract

### Background

Insomnia is a highly prevalent sleep disorder with substantial impacts on health and quality of life. Current treatment options have certain limitations, highlighting the need for novel therapeutic approaches. Transcutaneous auricular vagus nerve stimulation (ta-VNS) is gaining recognition as a promising non-invasive neuromodulation technique for treating insomnia, but its efficacy has not been systematically evaluated. Therefore, this protocol outlines the framework for a systematic review and meta-analysis designed to comprehensively assess the effectiveness and safety of ta-VNS in treating primary insomnia.

### Methods

A thorough literature search will be carried out in Embase, PubMed, the Cochrane Library, PsycINFO, AMED, PEDro, Wangfang database, Chinese National Knowledge Infrastructure, and Chinese BioMedical Literature Database, covering publications from their inception until December 31, 2024. Randomized controlled trials evaluating ta-VNS for primary insomnia in adults will be included. Two independent reviewers will screen studies, gather data, and evaluate the risk of bias based on the Cochrane RoB 2.0 tool. Meta-analyses will be conducted where appropriate, and the quality of evidence will be evaluated using GRADE. Subgroup and sensitivity analyses are planned.

### Discussion

This systematic review and meta-analysis will deliver a thorough synthesis of existing evidence regarding the efficacy and safety of ta-VNS for insomnia, potentially informing clinical practice and future research directions.

relevant data from this study will be made available upon study completion.

**Funding:** The project received funding from the Zhejiang Province Public Welfare Technology Application Research (grant number: LTGY23H270003; recipient: HH) and Zhejiang Provincial Famous Traditional Chinese Medicine Experts Inheritance Studio Construction Project (grant number: GZS2021027; recipient: HH). The funders had no role in study design, data collection and analysis, decision to publish, or preparation of the manuscript.

**Competing interests:** The authors have declared that no competing interests exist.

## Trial registration

**PROSPERO registration number:** CRD42024529039.

## 1 Introduction

Insomnia is a widespread sleep disorder involving problems with starting or sustaining sleep, waking up early in the morning, and resulting daytime functional impairments. Insomnia significantly impacts global health. A comprehensive review [1] highlights that around 10% of adults experience insomnia disorders, while an additional 20% deal with intermittent insomnia symptoms. Women, seniors, and those facing economic challenges are particularly susceptible to insomnia [1]. This condition tends to be long-lasting, with a 40% recurrence rate over five years [1]. A recent epidemiological study conducted in Norway highlights a substantial prevalence of insomnia, particularly among individuals with mental disorders. The study found that 61.3% of females with comorbid depression and 83.3% with comorbid schizophrenia reported insomnia [2]. These studies underscore the extensive burden of insomnia and its strong association with both mental and physical health conditions worldwide. Moreover, insomnia often persists chronically, following a pattern of relapse and remission [3]. As such, insomnia represents a major public health problem globally.

The pathophysiology of insomnia involves a complex interplay of psychological, behavioral, and physiological factors that lead to hyperarousal and an inability to down-regulate wakefulness [4]. Current treatment approaches for insomnia include cognitive-behavioral therapy for insomnia (CBT-I) as first-line treatment [5], followed by pharmacological therapies such as benzodiazepines, non-benzodiazepine hypnotics, melatonin receptor agonists, and sedating antidepressants [6]. However, the efficacy of existing treatments remains suboptimal for many patients, with issues such as limited access to CBT-I, modest effect sizes, side effects, and concerns about long-term use of medications [5]. As a result, there is a clear need for novel therapeutic options for insomnia that are effective, safe, well-tolerated and easily accessible.

Transcutaneous auricular vagus nerve stimulation (ta-VNS) is now recognized as a promising neuromodulation technique for a range of neurological and psychiatric conditions [7].

By delivering electrical stimulation to the auricular branch of the vagus nerve in the ear, ta-VNS can modulate brain activity in regions involved in autonomic function, emotion regulation, and cortical excitability [8]. Compared to implantable VNS, ta-VNS provides a noninvasive, convenient and low-risk alternative. ta-VNS has shown promising results in treating insomnia. In recent years, ta-VNS has become a potential non-invasive neuromodulation method for the treatment of insomnia, given its ability to modulate the autonomic nervous system and functional connectivity specific cortex between specific brain regions [9, 10]. Increasing clinical trials involving ta-VNS for treating insomnia have shown promising results. For instance, a randomized clinical trial [11] demonstrated that ta-VNS significantly improved sleep quality, reduced fatigue, and alleviated depression and anxiety symptoms in insomnia patients. Another study [12] found that ta-VNS significantly decreased the Pittsburgh Sleep Quality Index (PSQI) scores in participants with insomnia, highlighting its efficacy in improving sleep quality. Furthermore, Zhang et al. [13] reported that ta-VNS modulates brain functional connectivity of medial prefrontal cortex, which is crucial for sleep regulation, suggesting a potential mechanism for its effectiveness in treating primary insomnia. Notably, when compared to other non-invasive neuromodulation techniques like transcranial magnetic stimulation (TMS), which has been studied as a possible treatment for insomnia, ta-VNS presents

several potential advantages. Unlike TMS, which requires specialized equipment and trained personnel, potentially limiting its accessibility, ta-VNS is more portable and allows for self-administration by patients at home after initial training, which could improve treatment adherence and lower healthcare costs. Additionally, ta-VNS has a more favorable side effect profile, with most reported adverse events being mild and transient. However, it should be noted that the evidence base for ta-VNS in insomnia treatment is still developing, and more large-scale, long-term studies are needed to fully establish its efficacy and safety profile relative to other neuromodulation techniques.

To date, the effects of ta-VNS on insomnia have not been systematically evaluated. A comprehensive synthesis of the existing evidence is needed to 1) clarify whether ta-VNS is beneficial for insomnia; 2) quantify the magnitude of effects across studies; 3) characterize the safety of the intervention. Nonetheless, to date, no systematic review (SR) or meta-analysis of ta-VNS for insomnia has been conducted. Therefore, this proposed SR and meta-analysis aims to address this key knowledge gap by rigorously collating and analyzing all available data from randomized controlled trials (RCTs) of ta-VNS in insomnia. The timely synthesis of existing trial data will shed new light on the therapeutic potential of ta-VNS, accelerating progress towards evidence-based neuromodulation treatments for the millions suffering from insomnia worldwide.

## 2 Methods

The reporting of this section adheres to the Preferred Reporting Items for Systematic Reviews and Meta-Analysis Protocols (PRISMA-P) guidelines [14] to guarantee thorough and transparent reporting.

### 2.1 Protocol registration

This SR and meta-analysis protocol has been pre-registered in the International Prospective Register of Systematic Reviews (PROSPERO) with the registration number CRD42024529039.

### 2.2 Eligibility criteria

The eligibility criteria for study selection are carefully designed based on the PICOS (i.e., Populations, Interventions, Comparators, Outcomes, Study designs) framework.

**2.2.1 Populations.** Eligible participants will be adult patients (minimum age of 18 years) clinically diagnosed with primary insomnia according to well-established diagnostic manuals, such as the Diagnostic and Statistical Manual of Mental Disorders (Fourth Edition (DSM-IV) [15] or Fifth Edition (DSM-5) [16], and the Third version of the International Classification of Sleep Disorders (ICSD-3) [17]. Individuals whose insomnia is secondary to or comorbid with other physical health conditions such as pain or neurodegenerative diseases, will be excluded.

**2.2.2 Interventions.** ta-VNS can be administered alone or together with the same therapy as the control group. There will be no restrictions on stimulation parameters, duration of ta-VNS, or treatment frequency.

**2.2.3 Comparators.** Studies comparing ta-VNS with placebo, sham ta-VNS, no treatment, waitlist control, or other active treatments will be eligible for inclusion.

**2.2.4 Outcomes.**

*(1) Primary outcome.* Primary outcome is sleep quality, which can be assessed using self-reported scales known for their reliability and validity (e.g., the Insomnia Severity Index (ISI) [18], PSQI [19]). If available, objective sleep metrics, such as total sleep time and sleep

efficiency (the ratio of time spent asleep to time spent in bed), will also be measured using sleep detection devices.

*(2) Secondary outcomes. 1) The proportion of recovered patients.* It can be evaluated based on specified criteria, such as Sieigel's Criteria or other measures categorizing patients' recovery as complete, marked, slight, or none based on improved hearing in specified insomnia-related scales.

*2) Psychological symptom severity.* The severity of psychological symptoms, such as anxiety and depression, will be measured using standardized questionnaires. For instance, the Self-Assessment Anxiety Scale (SAS) [20] for evaluating the anxiety severity, and the Hamilton Depression Scale (HAMD) for evaluating the depression severity [21].

*3) Quality of life.* Quality of life can be assessed by validated and reliable questionnaires like the 36-Item Short Form Health Survey (SF-36) [22].

*4) Adverse events associated with interventions.* The number of adverse events (AEs) linked to the interventions will be evaluated.

**2.2.5 Study designs.** Only RCTs will be considered eligible study designs. Additionally, we will only accept studies with at least 20 participants per group, as this is widely recognized as the minimum for meaningful statistical analysis in clinical trials. The language of publication will be limited to English and Chinese. Non-RCTs, cohort studies, case reports, and other observational studies without a control group will be excluded.

## 2.3 Literature search strategy

A thorough search for literature will be performed in these bibliographic databases: Embase, PubMed, the Cochrane Library, PsycINFO, AMED, PEDro, Wangfang database, Chinese National Knowledge Infrastructure, and Chinese BioMedical Literature Database, covering publications from their inception until December 31, 2024. The search strategy will combine MeSH terms and free-text words related to "transcutaneous auricular vagus nerve stimulation," "ta-VNS," "insomnia," and "sleep disorders." Additionally, using the snowballing method for literature search, we will manually the reference lists of pertinent review articles on this topic. Table 1 contains the search strategy used in PubMed. To ensure a comprehensive search across all databases, we have made specific adjustments to other databases' search strategy. For instance, in English databases, we have expanded our search to include not only the subject headings "insomnia" but also related terms (e.g., "sleep disorder" "difficulty sleeping", "trouble sleeping") in both the title and abstract fields. This approach allows us to identify studies that use various terminologies to describe insomnia, thereby enhancing the sensitivity of our search. In the Chinese databases (Wangfang, CNKI, and CBM), we used Chinese terms equivalent to 'insomnia', 'sleep disorder', and various forms of 'vagus nerve stimulation' to ensure we capture all relevant Chinese literature. These strategies are designed to be as inclusive as possible while maintaining specificity to our research question. The detailed search strategies for the remaining databases are uploaded in the S1 and S2 Files.

Moreover, an add-on search for ongoing or unpublished trials will be undertaken by comprehensively searching a total of 5 major clinical trial registries as follows: International Clinical Trial Registration Platform, ClinicalTrials.gov registry, Australian New Zealand Clinical Trials Registry, Chinese Clinical Trial Registry, ISRCTN registry.

**Table 1. Search strategy in PubMed.**

| No. | Search items |
|---|---|
| #1 | Randomized controlled trial [Publication type] |
| #2 | Controlled clinical trial [Publication type] |
| #3 | Randomized OR Randomised [Title/Abstract] |
| #4 | Clinical trials [MeSH] |
| #5 | Randomly [Title/Abstract] |
| #6 | Trial [Title/Abstract] |
| #7 | #1 OR #2 OR #3 OR #4 OR #5 OR #6 |
| #8 | Humans [MeSH] |
| #9 | #7 AND #8 |
| #10 | Insomnia [MeSH] |
| #11 | insomnia OR sleep disorder OR difficulty sleeping OR trouble sleeping [Title/Abstract] |
| #12 | #10 OR #11 |
| #13 | "taVNS"[Title/Abstract] OR "ta-VNS"[Title/Abstract] OR "VNS"[Title/Abstract] |
| #14 | ("vagus nerve"[MeSH] OR "vagus nerve"[Title/Abstract] OR "vagus"[Title/Abstract] OR "vagal"[Title/Abstract]) AND "stimulation"[Title/Abstract] AND "transcutaneous"[Title/Abstract] |
| #15 | #13 OR #14 |
| #16 | #9 AND #12 AND #15 |

## 2.4 Study screening and data collection

The titles and abstracts of the retrieved articles will be independently screened by two reviewers. Subsequently, they will read the full texts of potentially qualified studies to determine study selection. Disputes will be settled by consulting a third referee.

Data from the included RCTs will be collected into a standardized form, which records study characteristics (e.g., authors, country, year of publication), patient characteristics (e.g., sex, mean age, diagnostic criteria for insomnia), intervention details (e.g., stimulation parameters of ta-VNS, treatment duration), comparator details, outcomes, and adverse events. The PRISMA flowchart (Fig 1) presents the procedure for study selection in details.

## 2.5 Risk of bias assessment

Two raters will independently appraise the methodological quality of enrolled studies via the Cochrane risk of bias 2.0 tool [23]. This tool assesses five key items: the randomization process, deviations from intended interventions, missing outcome data, outcome measurement, and the selection of the reported result. Every item will be evaluated and categorized as having a low risk of bias, some concerns, or a high risk of bias. The overall risk of bias for each study will then be appraised by considering the evaluations of these individual domains. In cases of disagreement between raters, a consensus will be reached through discussion. If consensus cannot be achieved, a third reviewer will be consulted to make the final decision.

## 2.6 Heterogeneity analysis of and data synthesis

Cochran's Q test and the $I^2$ statistic will be utilized to appraise statistical heterogeneity. Nevertheless, the decision to pool these studies will also consider clinical heterogeneity. Key clinical characteristics that will be considered when assessing clinical heterogeneity include patient demographics (e.g., age, gender), insomnia severity and duration, comorbid conditions, and specific ta-VNS protocols (e.g., stimulation parameters, treatment duration). If studies are judged to be too clinically heterogeneous to produce meaningful summary effect estimates

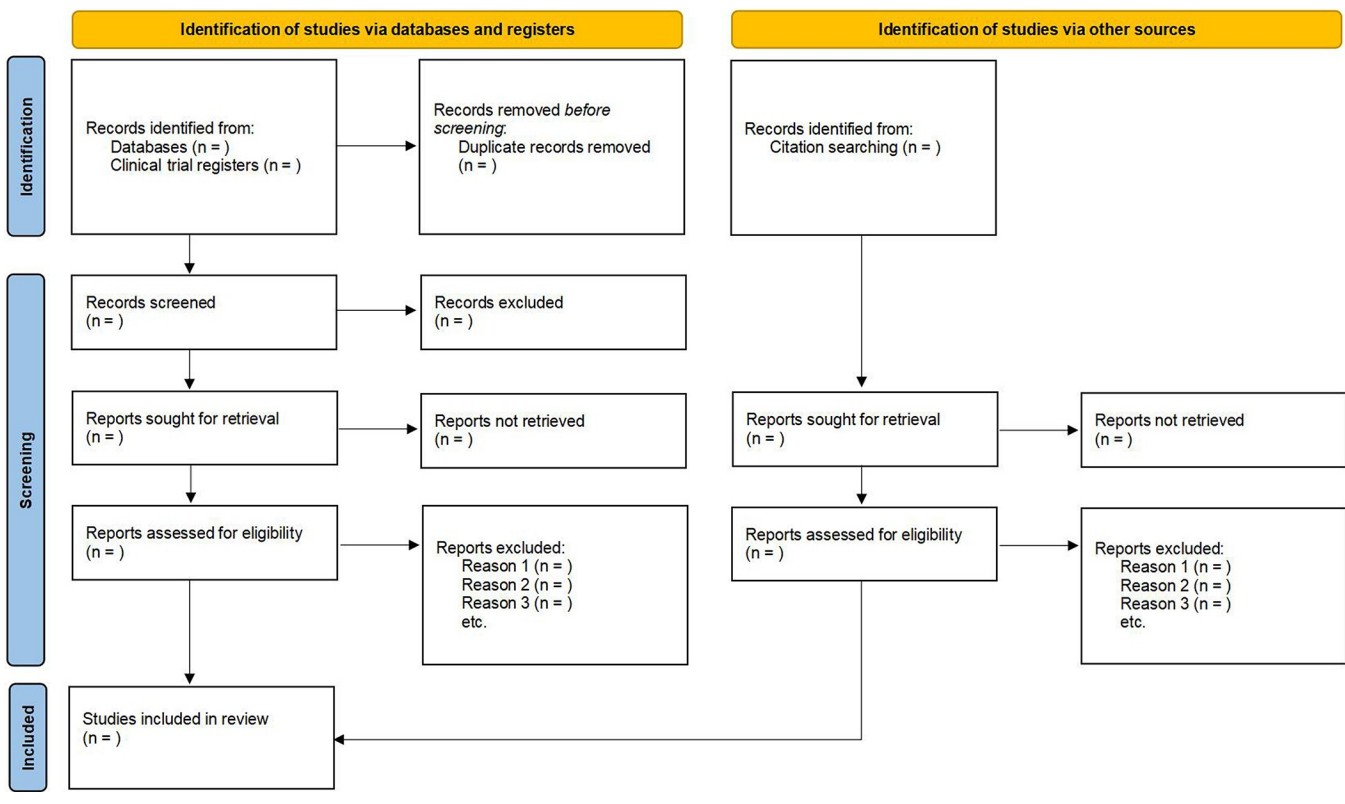

**Fig 1. Flow diagram of study selection.**

when pooled, we will not perform meta-analysis, even if there is no significant statistical heterogeneity. In such instances, a descriptive qualitative synthesis will be performed instead. If studies are found to be adequately homogeneous both clinically and statistically for performing meta-analysis, we will employ a fixed-effect model if the P-value from the Q test is greater than 0.10 and the $I^2$ value is 50% or less [24]. Should the p-value be less than 0.10 and the $I^2$ value exceed 50%, we will adopt a random-effect model. Data synthesis for the meta-analysis will be undertaken via RevMan version 5.4.1.

## 2.7 Subgroup and sensitivity analyses

If substantial heterogeneity is present, we will explore potential sources through subgroup analyses based on different participant characteristics, stimulation parameters, comparator interventions, and time points for primary outcomes. Sensitivity analyses will be conducted by excluding studies identified as having a high risk of bias.

## 2.8 Safety evaluation

The safety assessment of ta-VNS versus control interventions will involve pooling the number of AEs from the original studies through a meta-analytic approach.

## 2.9 Publication bias assessment

Funnel plots and Egger's test will be employed to evaluate publication bias, provided that the analysis includes a minimum of ten studies [25].

## 2.10 Evaluating the quality of evidence

The Grading of Recommendations, Assessment, Development, and Evaluation (GRADE) methodology will be utilized to evaluate the quality of evidence for each outcome [26]. This methodology takes into account several criteria, including imprecision, inconsistency of findings, indirectness of evidence, study limitations, and publication bias. According to these factors, the quality of evidence could be categorized as high, moderate, low, or very low.

## 3 Discussion

Insomnia is a common sleep disorder that can greatly affect an individual's quality of life, cognitive abilities, and mental well-being [27]. Current treatment options for insomnia, such as cognitive-behavioral therapy and pharmacological interventions, have limitations in terms of accessibility, adherence, and potential side effects [5]. ta-VNS has become an encouraging neuromodulation approach for treating insomnia. Regarding the therapeutic mechanisms of ta-VNS in treating insomnia, it is mainly linked to the modulation of autonomic nervous system function and brain activity in specific cortex or brain networks. For instance, a study [28] discovered that ta-VNS significantly altered the functional activity within the sensorimotor network and autonomic regulation, suggesting that these changes contribute to its efficacy in the treatment of insomnia [28]. Another study [13] indicated that ta-VNS decreases the functional connectivity (FC) between the medial prefrontal cortex (mPFC) and other brain cortices, including the dorsal anterior cingulate gyrus and the occipital cortex. This reduction in FC was correlated with improvements in sleep quality, indicating that ta-VNS helps regulate the default mode network and its interactions with other brain networks critical for sleep maintenance.

In recent years, the number of RCTs investigating the efficacy and safety of ta-VNS is increasing [11, 12]. Nonetheless, based on literature search, no systematic reviews or meta-analysis of ta-VNS for insomnia has been undertaken so far. The timely synthesis of current trial data will provide fresh insights into the therapeutic potential of ta-VNS, advancing the development of evidence-based neuromodulation treatments for the millions affected by insomnia globally. Consequently, this protocol outlines the framework for a subsequent SR and meta-analysis designed to comprehensive evaluate the therapeutic effect and safety of ta-VNS for the treatment of insomnia. While this meta protocol does not present final results of meta-analyses, we anticipate that our primary efficacy outcomes will focus on changes in validated insomnia scales (such as the ISI or PSQI) and objective sleep parameters (such as total sleep time and sleep efficiency) following ta-VNS treatment. We expect to conduct meta-analyses on these outcomes, which will provide quantitative estimates of the overall effect of ta-VNS on insomnia symptoms. These findings will be crucial in determining the clinical significance of ta-VNS as a treatment for insomnia. Therefore, this SR and meta-analysis will yield findings that are significant for clinical practice and research. If ta-VNS is found to be beneficial and safe for the treatment of insomnia, it could provide a valuable alternative or complementary therapy for patients who do not respond to or cannot tolerate conventional treatments. The results may also guide future research by identifying optimal stimulation parameters and treatment durations, as well as highlighting potential subgroups of patients who may benefit most from ta-VNS. Apart from its innovative nature as the first SR and meta-analysis on this topic, the strengths of this research include the use of a rigorous methodology, adherence to PRISMA guidelines, and a comprehensive search strategy encompassing multiple databases and clinical trial registries. The inclusion of RCTs ensures a high level of evidence, and the assessment of risk of bias and quality of evidence using well-established tools (RoB 2 and GRADE) enhances the reliability and transparency of our findings.

### 3.1 Limitations

First, the general quality of evidence may be impacted by factors such as small sample sizes in individual studies and potential risk of bias. Therefore, we will consider studies with a minimum sample size of 20 participants per group as acceptable, as this is generally considered the lower limit for meaningful statistical analysis in clinical trials. We also recognize the need for future studies to include long-term follow-up assessments. The current literature on ta-VNS for insomnia predominantly focuses on short-term outcomes, and there is a lack of data on the sustained effects and long-term safety of this intervention. This limitation highlights an important area for future research to address. Second, since the members of the review group are only proficient in English and Chinese and are unable to afford translation costs for other languages, this study includes only RCTs published in English and Chinese, which may result in potential language bias. Third, potential limitations may arise from the heterogeneity of the included studies in terms of participant characteristics, stimulation parameters, and comparator interventions. If available, subgroup and sensitivity analyses will be employed to resolve these issues and offer a more comprehensive perspective on the efficacy of ta-VNS for insomnia.

## 4 Conclusions

In conclusion, this planned SR and meta-analysis will provide a comprehensive evaluation of the efficacy and safety of ta-VNS for the treatment of insomnia. The findings will contribute to the growing body of evidence on non-invasive neuromodulation methods for sleep disorders and inform clinical decision-making and future research directions.

## Supporting information

**S1 File. PRISMA-P (Preferred Reporting Items for Systematic review and Meta-Analysis Protocols) 2015 checklist: Recommended items to address in a systematic review protocol.** (PDF)

**S2 File. Search strategies in other databases.**
(PDF)

## Author Contributions

**Conceptualization:** Lianqiang Fang, Hantong Hu.

**Investigation:** Xingling Li.

**Writing – original draft:** Ting Yang, Yunhuo Cai.

**Writing – review & editing:** Xingling Li, Lianqiang Fang, Hantong Hu.

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
