## [Decision Letter · Decision Letter 0]

12 Aug 2024

PONE-D-24-23880Is transcutaneous auricular vagus nerve stimulation effective and safe for primary insomnia? A PRISMA-compliant protocol for a systematic review and meta-analysisPLOS ONE

Dear Dr. Hu,

Thank you for submitting your manuscript to PLOS ONE. After careful consideration, we feel that it has merit but does not fully meet PLOS ONE’s publication criteria as it currently stands. Therefore, we invite you to submit a revised version of the manuscript that addresses the points raised during the review process.

We look forward to receiving your revised manuscript.

Kind regards,

Jaspinder Kaur, MD

Academic Editor

PLOS ONE

Reviewers' comments:

Reviewer's Responses to Questions

**Comments to the Author**

1. Does the manuscript provide a valid rationale for the proposed study, with clearly identified and justified research questions?

Reviewer #1: Yes

Reviewer #2: Yes

2. Is the protocol technically sound and planned in a manner that will lead to a meaningful outcome and allow testing the stated hypotheses?

Reviewer #1: Yes

Reviewer #2: Yes

3. Is the methodology feasible and described in sufficient detail to allow the work to be replicable?

Reviewer #1: Yes

Reviewer #2: Yes

4. Have the authors described where all data underlying the findings will be made available when the study is complete?

Reviewer #1: Yes

Reviewer #2: Yes

5. Is the manuscript presented in an intelligible fashion and written in standard English?

Reviewer #1: Yes

Reviewer #2: Yes

6. Review Comments to the Author

You may also provide optional suggestions and comments to authors that they might find helpful in planning their study.

Reviewer #1: Thank you for the opportunity to review this manuscript. I would like to commend the authors for their thorough investigation into the efficacy and safety of transcutaneous auricular vagus nerve stimulation (ta-VNS) for primary insomnia through a systematic review and meta-analysis. I have the following comments:

1. While the Introduction mentions ta-VNS as a noninvasive alternative to implantable VNS, it doesn't offer a detailed comparison with other non-invasive treatments for insomnia. Consider including a comparative analysis of ta-VNS with other non-invasive neuromodulation techniques such as transcranial magnetic stimulation (TMS) which has been explored as a pontential treatment for insomnia, while still a relatively novel approach some studies have shown promising results. If possible, highlight the advantages and potential limitations of ta-VNS compared to these treatments.

2. In section 2.3 Literature Search Strategy: It would be helpful to add a brief summary or highlight the key points from the search strategies of other databases within the main text. This summary could mention any unique adjustments made for specific databases and ensure that readers understand the comprehensiveness of the search strategy without referring to the supplementary file.

3. In Section: 2.5 Risk of Bias Assessment: The manuscript outlines the use of the Cochrane risk of bias 2.0 tool effectively. However, it could benefit from a brief mention of how disagreements between raters will be handled.

4. In Section: 2.6 Heterogeneity Analysis and Data Synthesis : Please add a brief sentence specifying some key clinical characteristics that will be considered when assessing clinical heterogeneity.

5. The Discussion provides a good foundation for the rationale and anticipated impact of the study, but could benefit from adding a clear and concise summary of the primary efficacy outcomes or any significant statistical findings derived from the meta-analysis.

6. In the Limitations section please specify the minimum sample size considered acceptable.

7. In the Limitation section Consider highlighting the need for future studies to include long-term follow-up to better understand the sustained effects and safety of ta-VNS.

Reviewer #2: I would like to congratulate the authors for their work and study design. The methodology is sound and adheres to high standards. The protocol is well written.

7. PLOS authors have the option to publish the peer review history of their article (what does this mean?). If published, this will include your full peer review and any attached files.

Reviewer #1: **Yes: **Mohsin Raza

Reviewer #2: **Yes: **Abbas Alshami

---

## [Author Response · Author response to Decision Letter 0]

8 Oct 2024

Point-by-point response to Reviewer 1's Comments

1.While the Introduction mentions ta-VNS as a noninvasive alternative to implantable VNS, it doesn't offer a detailed comparison with other non-invasive treatments for insomnia. Consider including a comparative analysis of ta-VNS with other non-invasive neuromodulation techniques such as transcranial magnetic stimulation (TMS) which has been explored as a pontential treatment for insomnia, while still a relatively novel approach some studies have shown promising results. If possible, highlight the advantages and potential limitations of ta-VNS compared to these treatments.

Response: 

Thank you for this insightful comment. We have addressed this by adding a comparative analysis of ta-VNS with transcranial magnetic stimulation (TMS) in the revised Introduction section (page 4, lines 104-114 in the clean version of the revised manuscipt; or you can directly check the revised manuscript with tracked changes) . We have highlighted the potential advantages of ta-VNS, such as its portability and potential for self-administration, as well as its favorable side effect profile. We have acknowledged that the evidence base for ta-VNS is still developing and that more research is needed to fully establish its comparative efficacy and safety. This addition provides a more comprehensive context for the potential of ta-VNS as a treatment for insomnia. 

For your earliest convenience, you can read the corresponding revisions in the revised manuscript (page 8, lines 197-198) as follows：

“Notably, when compared to other non-invasive neuromodulation techniques like transcranial magnetic stimulation (TMS), which has been studied as a possible treatment for insomnia, ta-VNS presents several potential advantages. Unlike TMS, which requires specialized equipment and trained personnel, potentially limiting its accessibility, ta-VNS is more portable and allows for self-administration by patients at home after initial training, which could improve treatment adherence and lower healthcare costs. Additionally, ta-VNS has a more favorable side effect profile, with most reported adverse events being mild and transient. However, it should be noted that the evidence base for ta-VNS in insomnia treatment is still developing, and more large-scale, long-term studies are needed to fully establish its efficacy and safety profile relative to other neuromodulation techniques.” 

2. In section 2.3 Literature Search Strategy: It would be helpful to add a brief summary or highlight the key points from the search strategies of other databases within the main text. This summary could mention any unique adjustments made for specific databases and ensure that readers understand the comprehensiveness of the search strategy without referring to the supplementary file.

Response: 

We appreciate this suggestion to improve the clarity and comprehensiveness of our search strategy description. We have added a brief summary in the main text that highlights key points from the search strategies for other databases. This summary mentions specific adjustments made for the remaining English databases and the Chinese databases (Wangfang, CNKI, and CBM). This addition provides readers with a clear understanding of our comprehensive approach without needing to refer to the supplementary file

For your earliest convenience, you can read the corresponding revisions in the revised manuscript (pages 7-8; lines 200-211 in the clean version of the revised manuscipt; or you can directly check the revised manuscript with tracked changes) as follows：

“Table 1 contains the search strategy used in PubMed. To ensure a comprehensive search across all databases, we have made specific adjustments to other databases’ search strategy. For instance, in English databases, we have expanded our search to include not only the subject headings "insomnia" but also related terms (e.g., "sleep disorder" "difficulty sleeping", "trouble sleeping") in both the title and abstract fields. This approach allows us to identify studies that use various terminologies to describe insomnia, thereby enhancing the sensitivity of our search. In the Chinese databases (Wangfang, CNKI, and CBM), we used Chinese terms equivalent to 'insomnia', 'sleep disorder', and various forms of 'vagus nerve stimulation' to ensure we capture all relevant Chinese literature. These strategies are designed to be as inclusive as possible while maintaining specificity to our research question.” 

3. In Section: 2.5 Risk of Bias Assessment: The manuscript outlines the use of the Cochrane risk of bias 2.0 tool effectively. However, it could benefit from a brief mention of how disagreements between raters will be handled.

Response: 

Thank you for pointing out this oversight. We have added some sentences in the revised manuscript (page 8, lines 237-239) to address how disagreements between raters will be handled. Specifically, we have stated that in cases of disagreement, consensus will be reached through discussion, and if consensus cannot be achieved, a third reviewer will be consulted to make the final decision. This addition ensures transparency in our methodology for risk of bias assessment.

4.In Section: 2.6 Heterogeneity Analysis and Data Synthesis: Please add a brief sentence specifying some key clinical characteristics that will be considered when assessing clinical heterogeneity.

Response: We appreciate this suggestion to provide more detail on our heterogeneity assessment. In the revised manuscript, we have added a sentence specifying key clinical characteristics that will be considered when assessing clinical heterogeneity. These include patient demographics (e.g., age, gender), insomnia severity and duration, comorbid conditions, specific ta-VNS protocols (e.g., stimulation parameters, treatment duration), and outcome measures used. This addition provides readers with a clearer understanding of how we will evaluate clinical heterogeneity across studies.

For your earliest convenience, you can read the corresponding revisions in the revised manuscript (page 9, lines 243-247) as follows：

“Nevertheless, the decision to pool these studies will also consider clinical heterogeneity., including variations in patient baseline characteristics and ta-VNS treatment protocols across the included RCTs. Key clinical characteristics that will be considered when assessing clinical heterogeneity include patient demographics (e.g., age, gender), insomnia severity and duration, comorbid conditions, and specific ta-VNS protocols (e.g., stimulation parameters, treatment duration).”

5.The Discussion provides a good foundation for the rationale and anticipated impact of the study but could benefit from adding a clear and concise summary of the primary efficacy outcomes or any significant statistical findings derived from the meta-analysis.

Response: Thank you for this suggestion to enhance the Discussion section. We have added a paragraph that outlines our anticipated primary efficacy outcomes, focusing on changes in validated insomnia scales and objective sleep parameters. We have explained that we expect to conduct meta-analyses on these outcomes to provide quantitative estimates of the overall effect of ta-VNS on insomnia symptoms. We have also emphasized the importance of these findings in determining the clinical significance of ta-VNS as a treatment for insomnia. This addition provides a clearer picture of the expected outcomes and their potential impact.

For your earliest convenience, you can read the corresponding revisions in the revised manuscript (page 11, lines 301-316) as follows:

“Consequently, this protocol outlines the framework for a subsequent SR and meta-analysis designed to comprehensive evaluate the therapeutic effect and safety of ta-VNS for the treatment of insomnia. While this meta protocol does not present final results of meta-analyses, we anticipate that our primary efficacy outcomes will focus on changes in validated insomnia scales (such as the ISI or PSQI) and objective sleep parameters (such as total sleep time and sleep efficiency) following ta-VNS treatment. We expect to conduct meta-analyses on these outcomes, which will provide quantitative estimates of the overall effect of ta-VNS on insomnia symptoms. These findings will be crucial in determining the clinical significance of ta-VNS as a treatment for insomnia. Therefore, this SR and meta-analysis will yield findings that are significant for clinical practice and research. If ta-VNS is found to be beneficial and safe for the treatment of insomnia, it could provide a valuable alternative or complementary therapy for patients who do not respond to or cannot tolerate conventional treatments. The results may also guide future research by identifying optimal stimulation parameters and treatment durations, as well as highlighting potential subgroups of patients who may benefit most from ta-VNS.” 

6.In the Limitations section please specify the minimum sample size considered acceptable.

Response: 

We appreciate this suggestion to provide more specificity in our Limitations section. We have added a sentence stating that we will consider studies with a minimum sample size of 20 participants per group as acceptable. We have explained that this is generally considered the lower limit for meaningful statistical analysis in clinical trials, while acknowledging that larger sample sizes would provide more robust evidence. This addition provides clarity on our criteria for study inclusion and the potential limitations of smaller studies.

7.In the Limitation section Consider highlighting the need for future studies to include long-term follow-up to better understand the sustained effects and safety of ta-VNS.

Response: 

Thank you for this important suggestion. We have added a paragraph in the Limitations section (page 11-12, lines 326-333) highlighting the need for future studies to include long-term follow-up assessments. We have acknowledged that the current literature on ta-VNS for insomnia predominantly focuses on short-term outcomes, and there is a lack of data on the sustained effects and long-term safety of this intervention. We have emphasized that this limitation highlights an important area for future research to address. This addition provides a more comprehensive discussion of the current limitations in the field and directions for future research.

For your earliest convenience, you can read the corresponding revisions in the revised manuscript as follows:

“First, the general quality of evidence may be impacted by factors such as small sample sizes in individual studies and potential risk of bias. Therefore, we will consider studies with a minimum sample size of 20 participants per group as acceptable, as this is generally considered the lower limit for meaningful statistical analysis in clinical trials. We also recognize the need for future studies to include long-term follow-up assessments. The current literature on ta-VNS for insomnia predominantly focuses on short-term outcomes, and there is a lack of data on the sustained effects and long-term safety of this intervention. This limitation highlights an important area for future research to address.” 

Point-by-point response to Reviewer 2's Comments

1. I would like to congratulate the authors for their work and study design. The methodology is sound and adheres to high standards. The protocol is well written.

Response: 

We are deeply grateful for your positive and encouraging comments on our study protocol. We have put considerable effort into developing a robust methodology that adheres to high standards in systematic review protocols, and we are pleased that this is reflected in our writing.

Your feedback affirms that we are on the right track with our approach. It motivates us to maintain this level of rigor throughout the execution of our systematic review and meta-analysis. We believe that a well-designed protocol is crucial for conducting a high-quality systematic review, and your comments reassure us that we have laid a strong foundation for our future study.

We hope that our revisions and explanations adequately address the reviewers' insightful comments. Thank you again for considering our manuscript and providing such thoughtful feedback to improve the work. 

Yours Sincerely, 

Hantong Hu, M.D

---

## [Decision Letter · Decision Letter 1]

18 Oct 2024

Is transcutaneous auricular vagus nerve stimulation effective and safe for primary insomnia? A PRISMA-compliant protocol for a systematic review and meta-analysis

PONE-D-24-23880R1

Dear Dr. Hu,

We’re pleased to inform you that your manuscript has been judged scientifically suitable for publication and will be formally accepted for publication once it meets all outstanding technical requirements.

Kind regards,

Yung-Hsiang Chen, Ph.D.

Academic Editor

PLOS ONE

Additional Editor Comments (optional):

Congratulations on the acceptance of your manuscript, and thank you for your interest in submitting your work to PLOS ONE.

Reviewers' comments:

Reviewer's Responses to Questions

**Comments to the Author**

1. Does the manuscript provide a valid rationale for the proposed study, with clearly identified and justified research questions?

Reviewer #1: Yes

2. Is the protocol technically sound and planned in a manner that will lead to a meaningful outcome and allow testing the stated hypotheses?

Reviewer #1: Yes

3. Is the methodology feasible and described in sufficient detail to allow the work to be replicable?

Reviewer #1: Yes

4. Have the authors described where all data underlying the findings will be made available when the study is complete?

Reviewer #1: Yes

5. Is the manuscript presented in an intelligible fashion and written in standard English?

Reviewer #1: Yes

6. Review Comments to the Author

You may also provide optional suggestions and comments to authors that they might find helpful in planning their study.

Reviewer #1: After reviewing both the original and revised manuscripts, I am satisfied that the authors have thoroughly addressed all the key points raised. They have successfully incorporated a comparative analysis of ta-VNS with other non-invasive treatments, improved the clarity of the literature search strategy, and included details on risk of bias and clinical heterogeneity. The discussion now offers a clearer focus on primary outcomes, and the limitations section appropriately covers sample size considerations and the need for long-term follow-up. Overall, the revisions align well with the feedback provided

7. PLOS authors have the option to publish the peer review history of their article (what does this mean?). If published, this will include your full peer review and any attached files.

Reviewer #1: **Yes: **Mohsin Raza

---

## [Editor Report · Acceptance letter]

29 Oct 2024

PONE-D-24-23880R1 

PLOS ONE

Dear Dr. Hu, 

I'm pleased to inform you that your manuscript has been deemed suitable for publication in PLOS ONE. Congratulations! Your manuscript is now being handed over to our production team.

Kind regards, 

on behalf of

Dr. Yung-Hsiang Chen 

Academic Editor

PLOS ONE